# Fatty Acids-Enriched Fractions of *Hermetia illucens* (Black Soldier Fly) Larvae Fat Can Combat MDR Pathogenic Fish Bacteria *Aeromonas* spp.

**DOI:** 10.3390/ijms22168829

**Published:** 2021-08-17

**Authors:** Heakal Mohamed, Elena Marusich, Yuriy Afanasev, Sergey Leonov

**Affiliations:** 1Moscow Institute of Physics and Technology, School of Biological and Medical Physics, 141700 Dolgoprudny, Russia; m.heakal@phystech.edu (H.M.); yurii.afanasev@phystech.edu (Y.A.); 2Institute of Cell Biophysics, Russian Academy of Sciences, 142290 Pushchino, Russia

**Keywords:** *H. illucens*, *Aeromonas* spp., multi-drug resistance, sequential extraction, free fatty acids, antimicrobial activity

## Abstract

*Aeromonas* spp. cause many diseases in aquaculture habitats. *Hermetia illucens* (Hi) larvae were used as feed-in aquacultures and in eradicating pathogenic fish bacteria. In the present study, we applied consecutive extractions of the same biomass of BSFL fat using the acidic water–methanol solution. The major constituents of the sequential extracts (SEs) were free fatty acids (FFAs), and fatty acids derivatives as identified by gas chromatography spectrometry (GC-MS). Our improved procedure enabled gradual enrichment in the unsaturated fatty acids (USFAs) content in our SEs. The present study aimed to compare the composition and antimicrobial properties of SEs. Among actual fish pathogens, *A. hydrophila* and *A. salmonicida* demonstrated multiple drug resistance (MDR) against different recommended standard antibiotics: *A. salmonicida* was resistant to six, while *A. hydrophila* was resistant to four antibiotics from ten used in the present study. For the first time, we demonstrated the high dose-dependent antibacterial activity of each SE against *Aeromonas* spp., especially MDR *A. salmonicida*. The bacteriostatic and bactericidal (MIC/MBC) activity of SEs was significantly enhanced through the sequential extractions. The third sequential extract (AWME3) possessed the highest activity against *Aeromonas* spp.: inhibition zone diameters were in the range (21.47 ± 0.14–20.83 ± 0.22 mm) at a concentration of 40 mg/mL, MIC values ranged between 0.09 and 0.38 mg/mL for *A. hydrophila* and *A. salmonicida*, respectively. AWME3 MBC values recorded 0.19 and 0.38 mg/mL, while MIC_50_ values were 0.065 ± 0.004 and 0.22 ± 0.005 mg/mL against *A. hydrophila* and *A. salmonicida*, respectively. Thus, the larvae fat from *Hermitia illucens* may serve as an excellent reservoir of bioactive molecules with good capacity to eradicate the multidrug-resistant bacteria, having promising potential for practical application in the aquaculture field.

## 1. Introduction

Aquaculture is a food-production sector that permanently grows worldwide. It produced nearly 74 million tons of fish, and approximately 45% of the global production of fish-based food, in 2014. Because of the increasing demand for food protein and the stagnation of wild catch, fish production by aquaculture is expanding, with a growth rate exceeding 8% per year [1]. Aquaculture confronts several problems, including the corruption of natural ecosystems, water contamination, biological pollution, and the emergence of diverse fish diseases.

The use of antimicrobials in fish farming mirrors the fast aquaculture development worldwide. The intensification of aquaculture to achieve market demands could increase infectious diseases by pathogenic bacteria. Consequently, antimicrobials control emerging infectious diseases, although their use must follow the rules and local regulations of the country where they are explored. Bacterial fish diseases are usually treated with antibiotics which, over time, lead to bacterial resistance [2]. The selection of resistant bacteria is closely associated with antibiotic treatments, the co-selection of resistance genes, and cross-resistance processes. The aquatic environment (freshwater and marine) can serve as a reservoir of resistant bacteria and genes encoding antimicrobial resistance [3]. Aquaculture is yet another environmental gateway to the development and globalization of antimicrobial resistance. The development and spread of resistant bacteria combined with antimicrobial residues in the environment represent dangerous risks to public health [4]. Resistant bacteria from fish farming are a severe concern because humans can acquire them with handling or food chain, representing a public health problem.

*Aeromonas* spp. are the most common bacteria in freshwater and saline habitats frequently associated with severe infections in cultured fish species [5]. The *Aeromonas* genus comprises a group of Gram-negative anaerobic bacteria found in soil and water. Besides, it can be associated to various infectious diseases found in animals and humans [6] classified into two groups: mobile mesophilic (optimal growth between 35 and 37 °C), associated to various human diseases and psychrophilic, non-mobile (optimal growth between 22 and 25 °C), which can infect both fish and reptiles [7]. 

The main species associated with infections in fish are *Aeromonas hydrophila*, *A. sobria*, *A. salmonicida*, and *A. veronii* [6,8]. The stress generated by inadequate management can generate a trigger for the appearance of infections caused by *Aeromonas* sp. in fish production [9]. Consequently, the use of antibiotics is the alternative for controlling the disease. 

*A. hydrophila* is a Gram-negative rod-shaped bacterium is the leading causative agent of *Aeromonas septicemia* known as tail and fin rot. This bacterium causes severe infections for fish species, including loach (*Misgurnus anguillicaudatus*), channel catfish (*Ictalurus punctatus*), and common carp (*Cyprinus carpio*); additionally, it can infect some marine fish species [10]. Aeromonads produce different types of toxins, such as cytotoxic enterotoxin, cytotonic enterotoxin, aerolysin, and hemolysin. They also secrete several extracellular enzymes such as lipase, elastase, gelatinase, protease, and DNAse. The secretion system, flagella system, and biofilm components are associated with the potential virulence of *Aeromonas* spp. [11,12]. The symptoms of these infections include ulcers, abdominal distension, accumulation of fluid, anemia, and hemorrhaging, resulting in mass mortality in fishes around the world [1,10,13]. In farming practices, various methods are applied to reduce the existence of *Aeromonas* spp. infection such as vaccination, water chlorination, and antibiotic chemotherapy. It was reported that the excess use of antibiotics accelerates the emergence of multidrug-resistant strains of *A. hydrophila* in aquaculture habitats [5].

*A. salmonicida* subsp. *salmonicida* is a non-motile Gram-Negative bacterium that causes furunculosis in salmonid fish. Therefore, it infects zebrafish in the aquatic environment, and it can multiply rapidly within a few hours of the post-infection [14]. Bacteria can be resistant to certain antimicrobials naturally or by horizontal gene transfer [15]. This pathogen causes hemorrhagic sepsis, ulcerative lesions, pointed bleeding, and death [16].

Multidrug resistance of food-borne bacteria is a paramount public health concern. The overuse of antimicrobials in aquaculture may cause antimicrobial contamination in coastal water environments [17]. The intrinsic resistance of the *Aeromonas* genus to beta-lactams antibiotics is associated with a chromosomal beta-lactamase expression or activation of efflux pumps [18]. Genes that confer resistance to a broad spectrum of beta-lactams antibiotics were identified in the genus. In particular, almost 10.5% of *Aeromonas* sp., isolated from an estuary, possessed beta-lactamases genes, especially the gene *bla*TEM [19]. Extended—Spectrum Beta—Lactamases (ESBL) are encoded on mobile genetic elements, such as plasmids, transposons, and integrons, which also harbor resistance genes to other classes of antimicrobials [20]. Strains producing ESBL can resist treatment and increase the morbidity of diseases associated with bacterial infections. Resistance to at least two classes of antimicrobials observed in 59% of untreated water isolates [21]. Moreover, the presence of the *tet* gene encoding resistance to tetracycline in 10%, and the *bla* gene in approximately 29% of isolates was observed. The *tet* gene found in *Aeromonas* species is considered as a result of strong anthropogenic pressure on aquaculture due to the use of tetracycline as antimicrobial [22]. On the other hand, in Australia, where no antimicrobials are licensed for use in aquaculture, 100% of *Aeromonas* strains isolated from fish carried the *tet* gene [23]. Albeit *Aeromonas* sp. presents typical characteristics of antimicrobial susceptibility, resistance profiles vary between studies because of the specific traits of the strains or selective pressure of the environment [12]. That prompted the necessity of antibiotic resistance profiling of each laboratory strain before comparison with new testing antimicrobials.

The activation of efflux pumps is yet another important factor in the resistance of *Aeromonas* sp. is linked. The AheABC system, encoded by genes *Ahe*A, *Ahe*B, and *Ahe*C of *A. hydrophila,* is involved in the phenotype of multidrug resistance to cefuroxime, cefoperazone, erythromycin, lincomycin, pristinamycin, minocycline, trimethoprim, fusidic acid, and rifampin [24].

Except for a few strains and the species of *Aeromonas* spp. are described as resistant to ampicillin [1,13,25]. These bacteria are also resistant to penicillin and first-generation cephalosporin groups [10,26,27]. *Aeromonas* spp. can be susceptible to monobactams, carbapenems, third- and fourth-generation cephalosporins, aminoglycosides, and fluoroquinolones [28]. Ndi et al. [23] reported an increase in the resistance to beta-lactam antimicrobials (penicillins and derivatives, cephalosporins, carbapenems, and monobactams) by the presence of genes that code for the production of beta-lactamases [10]. Recently, the studies of antimicrobial profiles in *Aeromonas* spp. have increased due to the necessity of responsible use of antibiotics [29,30,31,32]. 

Among several insect species, the *Hermetia illucens* (Black Soldier Fly or BSF) has a promising economic role for aquafeed production. From a nutritional point of view, the BSF accumulates high amounts of proteins and lipids (307.5–588.0 g/Kg and 113.0–386.0 g/Kg, respectively [33,34]. In terms of fatty acid profile, the BSF is usually rich in saturated fatty acids (SFAs) and poor in polyunsaturated (PUFAs) ones [35], which are extremely important for fish [36]. PUFAs deficiencies during fish farming can cause a general decrease of fish health, poor growth, low feed efficiency, anemia, and high mortality [37,38,39]. Recently, some authors demonstrated that rearing BSF larvae on an organic substrate containing proper amounts of omega-3 fatty acids was a suitable procedure to improve the FAs profile of the insect biomass [40]. New ingredients to be introduced in aquafeeds must be carefully analyzed, since it is well established that different feed ingredients may have modulatory influences on the fish physiological responses and gut microbiota [41,42].

Several recent studies reported the antimicrobial activity of the larvae hemolymph and maggot extract as well as of secretions are promising for the development of new therapeutically valuable antimicrobials, particularly in defense against multi-resistant “superbugs” [43,44]. It was reported the high nutritional value of unsaturated oleic (18:1, n-9) and linoleic acids (18:2, n-6) from BSFL fat [40,45]. Besides, BSFL fat is rich in medium-chain lauric acid with known antimicrobial activity towards the disruption of the bacterial cell membrane [46]. Recently we demonstrated the substantial antibacterial potential of acidic water–methanol extracts (AWME) of *H. illucens* larvae fat against contemporary phytopathogens [47].

In the present study, we attempted to improve our extraction procedure by using three sequential extracts (SEs) of the same biomass of BSFL fat using the acidic water–methanol solution. The antimicrobial susceptibility of *A. hydrophila* and *A.* salmonicida to the panel of standard antibiotics and every SE was assessed. We demonstrated that the major extracted constituents were free fatty acids (FFAs) and fatty acids derivatives, and the sequential extraction continuously improved their antibacterial activity. Our study, for the first time, demonstrated the high dose-dependent bactericidal activity of each SEs against pathogenic fish bacteria, especially MDR *A. salmonicida.* The bacteriostatic and bactericidal (MIC/MBC) activity of SEs was significantly enhanced through the sequential extraction of the same sample of BSFL fat. Besides, via our improved procedure, we were able to gradually enrich the unsaturated fatty acids (USFAs) content in our SEs. Thus, the larvae fat from *Hermitia illucens* may serve as an excellent reservoir of bioactive molecules with good capacity to eradicate the multidrug-resistant bacteria, having promising potential for practical application in the aquaculture field.

## 2. Results

### 2.1. Antimicrobial Susceptibility Testing

*Aeromonas* spp. can acquire antimicrobial resistance mechanisms, it is a candidate for indicator bacteria to follow antimicrobial resistance dissemination in aquatic environments. To study the antimicrobial susceptibility, clinicians and epidemiologists/microbiologists have two different approaches: clinicians focus on the tryptic microorganism/antibiotic/host and others on the pair microorganism/antibiotic to recognize microorganisms as wild type or non-wild type, meaning for absence or presence of any acquired and mutational resistance mechanism to the drug in question. Thus, epidemiologists/microbiologists explored the evolution or emergence of bacterial populations displaying resistant traits, regardless of any therapeutic outcome. Unfortunately, to date, interpretation criteria for *Aeromonas* spp. for antimicrobial susceptibility tests are scarce in the literature. Epidemiological Cutoff Values (ECVs) or ECOFFs, the interpretive criteria called by Clinical and Laboratory Standards Institute (CLSI) and European Committee on Antimicrobial Susceptibility Testing (EUCAST), were obtained from different laboratories and represent the upper limit of the MIC data distribution of fully susceptible (wild type) strains. Among *Aeromonas*, ECVs are available only for the species *A. salmonicida* regarding florfenicol, trimethoprim-sulfadimethoxine, oxytetracycline, and oxolinic acid either MICs or for Inhibition Zone Diameter (IZD) obtained by disk diffusion and for gentamicin, erythromycin, and trimethoprim-sulfamethoxazole (only IZD) [48]. Whereas antibiotic susceptibility of clinical isolates of *Aeromonas* has been extensively studied, less is known about environmental strains and particularly those from fish. Therefore, we first tested the susceptibility of two major fish pathogens for IZD by disk diffusion assay.

The resistance of fish pathogenic strains *A. salmonisida* and *A. hydrophila* to the standard antimicrobial agents represented eight classes of commonly used antibiotics (Appendix A and Table 1) using the inhibition zone diameters measurements at 12 h and 24 h of incubation and categorized by the comparison with the measures of the CLSI breakpoints [48].

Ten antibiotics of eight different chemical groups used at concentration, recommended by production company including: (1) aminoglycosides—penicillin/streptomycin (P/S, 100 U/Ml–100 µg/mL/disk), kanamycin (K, 1000 µg/disk); gentamycin (G, 200 µg/mL/disk); (2) tetracyclines—doxycycline (DX, 100 µg/mL/disk); (3) phenicols—chloramphenicol (Ch, 1000 µg/mL/disk); (4) penicillin (P, 2U/disk); (5) glycopeptides—vancomycin (VA, 5 µg/disk), (6) macrolides—erythromycin (E, 60 µg/disk); (7) Rifamycins: Rifampicin (RD, 15 µg/disk), and (8) lipopeptides—colistin (10 µg/disk). Both tested *A. hydrophila* and *A. salmonicida* strains demonstrated bacteria multidrug-resistant capacity (MDR). *A. salmonicida* was more resistant than *A. hydrophila* against six different groups of antibiotics, including penicillin, vancomycin, erythromycin, doxycycline, rifampicin, and colistin. *A. hydrophila* was moderately sensitive to colistin (CT) and resistant to doxycycline (DOX). Of note, both strains were 100% resistant to penicillin and vancomycin without visible IZD zone around the disks antibiotics. The intermediate sensitivity to penicillin-streptomycin (P/S) was observed for both strains. Thus, *A. salmonicida* was resistant to 60% while *A. hydrophila* was resistant 50% of the total ten antibiotics used in this test. The measurement of IZD at 12 h and 24 h demonstrated the decrease by the time of zone inhibition diameter caused by various antimicrobial drugs. Interestingly, treatment with rifampicin, erythromycin, and chloramphenicol showed the sharpest decreasing IZD after 24 h incubation at *A. hydrophila* and *A. salmonicida* strains, as indicated in Table 1, Appendix A. 

### 2.2. Sequential Extraction

As described in our previous work, the bioactive compounds from AWME extract received using our developed protocol of BSFL fat extraction demonstrated a high level of antimicrobial activity against a broad spectrum of bacterial phytopathogens [47]. In attempt to improve the extraction procedure, we applied three rounds of sequential extractions of the same biomass of BSFL fat using the same protocol based on acidic water–methanol extraction solution (90:9:1 *v/v*%) in order to get the set of sequential extracts (SEs) named as AWME1, AWME2, and AWME3 (Figure 1). The total amount of bioactive compounds for each SE obtained 60 mg, 40 mg, and 30 mg for AWME1, AWME2, and AWME3, respectively. The yield of consecutive SEs corresponds to 2%, 1.33%, and 1% for AWME1, AWME2, and AWME3, respectively, out of the initial 3 g of larvae fat.

### 2.3. Antibacterial Susceptibility Testing of Aeromonas *spp.* to SEs

The antibacterial activity of SEs was determined separately for each extract using the inhibition zone diameters measurements at 12 h and 24 h of incubation. In addition, we assessed the mixture of three equal volumes of SEs adjusted to the same 40 mg/mL final concentration. IZD around the discs loaded with AWME1, AWME2, and AWME3 and a mixture of them are shown in Figure 2. 

Inhibition Zone Diameter ranged from 11.4 ± 0.14 mm to 12.72 ± 0.22 mm for AWME1, while AWME2 against *A. hydrophila*a and *A. salmonicida*, respectively, by 24 h of incubation at concentration 40 mg/mL. IZD caused by AWME1 was lower significant (** *p* < 0.0047) than other SEs (AWME2, AWME3) at 12 h of incubation time against *A. salmonicida* as shown in Figure 3. The inhibition zone diameter caused by AWME3 was significantly higher (**** *p* < 0.0001) than for all other sequential extracts after 12 h and 24 h of incubation, as illustrated in Figure 3. The AWME3 demonstrated the most significant inhibition zone with sizes 21.47 ± 0.14 mm, 20.83 ± 0.22 mm against *A. hydrophila* and A. salmonicida, respectively, by 24 h of incubation. The activity of the mixture of three extracts was lower than that of AWME3 alone. Notably, whereas the susceptibilities of *A. hydrophila* to AWME3 and doxycycline (the recommended positive control) were almost equal, the *A. salmonicida* was significantly (**** *p* < 0.0001) more susceptible to AWME3 compared to the positive control. Our data demonstrated that the sequential extraction procedure gradually enhanced the antimicrobial activity SEs against the pathogenic fish bacteria.

### 2.4. AWME3 Demonstrates Dose-Dependent Antimicrobial Activity

Various concentrations of AWME3 were tested against both *A. hydrophila* and *A. salmonicida* strains (Figure 4), and the resulted IZD values are presented in Figure 5.

Dose-dependent susceptibility of *A. hydrophila* and *A. salmonicida* was observed in the inhibition zone sizes of *A. hydrophila* and *A. salmonicida* strains when they were subjected to AWME3 for 12 h and 24 h with concentrations ranging between 1.88 and 40 mg/mL (Figure 5). Notably, *A. salmonicida* was more susceptible to AWM3 than *A. hydrophila*, while demonstrating more significant (** *p* < 0.001, **** *p* < 0.0001) than positive control IZD values at 15–40 mg/mL of concentrations on both 12 h and 24 h. AWME3 from BSFL fat proved its ability to inhibit the fish pathogenic bacteria *A. salmonicida* in a dose-dependent manner.

### 2.5. Antibacterial Susceptibility Testing of Aeromonas *spp.* to SEs by MIC/MBC Assay

No epidemiological cut-off values for *Aeromonas* are currently available at EUCAST to interpret Minimum Inhibitory Concentrations (MIC). While MIC is the lowest concentration of an antibacterial agent necessary to inhibit visible growth, minimum bactericidal concentration (MBC) is the minimum concentration of an antibacterial agent that results in bacterial death defined by the inability to re-culture bacteria. The closer the MIC is to the MBC, the more bactericidal the compound [49]. 

Antibacterial activity of AWME1, AWME2, AWME3, and a mixture of all three SEs displayed a broad spectrum of antimicrobial activities was determined by broth, the microdilution assay method to assess the (MICs) and minimal bactericidal concentration (MBCs) values against two Gram-negative fish pathogens as shown in Table 2. The results demonstrated the highest bactericidal potency (both in MICs and MBCs) of the AWME3 against both tested Gram-negative *Aeromonas* pathogens, gradually decreasing from the second (AWME2) to the first (AWME1) sequential extracts. The mixture of AWME1, AWME2, and AWME3 (MIX) found to be less potent than AWME3 alone, albeit more potent than either AWME1 or AWME2 alone. Both AWME3 and MIX were more bactericidal against *A. salmonicida* than against *A. hydrophila,* as indicated by their equal MIC and MBC values. Surprisingly, *A. salmonicida* appeared less susceptible to both AWME3 and MIX compared to *A. hydrophila,* as demonstrated by higher MIC values. These results further supported our hypothesis that AWME3 is the most active extract among all three SEs and can inhibit the pathogenic fish bacteria at the lowest dose.

Based on our findings, AWME3 was selected to further evaluate this extract as the most potential for antimicrobial capacity against Gram-negative bacterial fish pathogens.

### 2.6. The Half of MIC (MIC_50_) and Growth Curves of AWME3 against Aeromonas *spp.*

The mean values of the half of the minimal inhibition concentration (MIC_50_) as well as the range of values obtained are important parameters for reporting results of susceptibility testing when several isolates of a given species are tested. The MIC_50_ represents the MIC value at which ≥50% of the isolates in a test population are inhibited; it is equivalent to the median MIC value.

The AWME3 concentration was adjusted into the correct concentration by mixing antimicrobial stock with media. The adjusted antimicrobial is serially diluted into multiple wells to obtain a gradient. The microbes came from the same colony-forming unit at the correct concentration adjusted to 5 × 10^5^ CFU/mL (MIC_50_ were determined at 6 h, 12 h, and 24 h by measuring the turbidity (OD_600_) of tested strains [50] and are presented in Figure 6. 

Comparison of MIC_50_ values indicated that *A. salmonicida* strain demonstrated higher resistance (more than 18-fold increase of MIC_50_) to DOX than *A. hydrophila* strain at all time intervals tested. In contrast, both strains were less different (only a 2–3-fold difference of MIC_50_ values) in susceptibility to AWME3 at any time point.

The effect of AWME3 and doxycycline (the recommended antibiotic positive control, DOX) on the number of live cells in a bacterial population over 24 h period of *A. hydrophila* and *A. salmonicida* division measured by turbidimetric assay is illustrated in Figure 7. The treatment with AWME3 at the concentrations ≤24 µg/mL extended the lag phase of *A. hydrophila* growth to 4 h, whereas in the presence of DOX at the concentrations ≤0.125 µg/mL it was only two hours. The lag phase of *A. salmonicida* under treatment with both AWME3 at the concentrations ≤190 µg/mL and DOX at concentrations ≤0.78 µg/mL were two hours. There was no exponential (or log) phase of *A. hydrophila* division under treatment with both AWME3 at the concentrations ≥190 µg/mL and DOX at concentrations ≥0.250 µg/mL, whereas *A. salmonicida* did not grow under both AWME3 and DOX treatments at the concentrations ≥380 µg/mL and ≥6.25 µg/mL, respectively. Notably, the AWME3 treatment at the concentration 95 µg/mL delayed the subtle *A. hydrophila* division up to 12 h, while the slow division of *A. salmonicida* began only at 8 h. In contrast, the DOX treatment at any concentrations ≤ of 0.125 µg/mL did not delay the time of *A.hydrophila* log phase appearance, while delaying the shallow logarithmic growth of *A. salmonicida* up to 10 h at concentrations within the range 1.56–3.12 µg/mL only. Under AWME3 treatment at the concentrations ≤12 µg/mL the stationary phase of *A. hydrophila* growth began to appear starting from 20 h with further shallow progression, while AWME3 at the concentrations ≤95 µg/mL induced less expression of the stationary phase of *A.salmonicida* division and the delay of the phase appearance up to 18 h at the concentration 190 µg/mL. DOX induced stationary phase of *A. hydrophila growth* appearance started from 8 h at the concentration 0.125 µg/mL and then gradually decreased from 18 h down to 8 h within the decrement of the concentrations from 0.063 µg/mL to 0.008 µg/mL. The same antibiotic persuaded the appearance of stationary-like *A. salmonicida* division starting from twenty hours at concentrations 1.56–3.12 µg/mL and less obvious stationary-like division from 12 h with further shallow progression at the concentration range 0.19–0.78 µg/mL.

These results demonstrate that the AWME3 from BSFL fat can prevent and sustainably inhibit the proliferation and growth of the fish pathogens, including antibiotic-resistant *A. salmonicida* strain, as early as 6 h of incubation, thus potentially being used as an antibacterial agent.

### 2.7. GC-MS Analysis of the Sequential Extracts from BSFL Fat

The GC-MS analysis identified 28–33 organic compounds in the AWMEs of larvae fat (Table 4). The chemical profile of these compounds was determined based on the National Institute of Standards and Technology (NIST, Gaithersburg, MD, USA) database. After comparing the mass spectrum of the unknown AWME components with the range of the known chemicals from NIST library, the similarity of GC-MS spectrums more than 70% considered as the main criteria for that selection.

The GC–MS chromatogram of three sequential extracts (SEs) of BSFL fat showed a slight difference in its composition (see Appendix A). GC-MS data revealed that the first extract (AWME1) contained 28 peaks, while the second (AWME2) and third (AWME3) contained 31 and 33 peaks, respectively. The peaks were identified by comparing the mass spectra with NIST-8 library database (Appendix A). The chemical structure, retention time, molecular formula, molecular weight, and concentration (peak area %) are presented in Appendix A, respectively. The GC-MS analysis showed the significant constituents of free fatty acids (FFAs) and their derivatives into each SEs of *H. illucens* larva fat. Among FFAs, the content of saturated fatty acids (SFAs) dominates two folds by the concentration, compared to the unsaturated fatty acids (USFA). The content of SFAs in AWME1, AWME2, and AWME3 59.2%, 51.09%, and 51.32%, while the USFAs was 26.05%, 27.42%, and 29.64%, respectively (Table 4). The major FFAs were n-hexadecanoic acid (palmitic acid), octadec-9-anoic acid (oleic acid), dodecanoic acid (lauaric acid), tetradecanoic acid (myristic acid), and octadecanoic acid (stearic acid). The AWME3 as the most potent antibacterial agent in inhibiting the pathogenic fish bacteria was enriched in cis-oleic acid (C18:1, 26.28%) and glycerol (C3:0, 7.87%) compared to other SEs. AWME3 comprises several compounds with one or more cis-double bonds such as cis-oleic acid; 9,12-hexadecadienoic acid methyl ester; cis-9-hexadecenal; 2,4-dodecadienal, (E, E); lauric acid beta-monoglyceride; oxiraneundecanoic acid, 3-pentyl- methyl ester, cis- were not found in AWME1 and AWME2 (Table 4). GC-MS result showed that AWME3 possessed the highest amount of USFAs, compounds with cis-double bonds, and glycerol suggested to be the most potent antibacterial among the other SEs constituents extracted from the BSFL fat.

## 3. Discussion

As we reported earlier regarding the anti-phytopathogenic properties of larvae fat extracts [47], the black soldier fly (BSF) *H. illucens* is a valuable natural source of biologically active compounds exceptionally enriched in the lipids among other insects [51]. The larvae’s lipid content is mainly rich in lauric, myristic, palmitic, oleic, capric, linoleic, and other medium-chain fatty acids, where myristic acid has a broad spectrum of antibacterial effects [52], larvicidal, and repellent activities [53]. Choi and Jiang [54] reported the activity of hexanedioic acid extracted from BSFL against Gram-positive and Gram-negative bacteria. The composition of lipids varies depending on the method of larvae processing that can yield various fatty acids (FAs) profiles [34,47].

In an attempt to improve the extraction method of larvae processing, in the present study, we implemented three consecutive extractions, obtaining AWME1, AWME2, and AWME3 extracts from *Hermetia illucens* (Hi) larvae fat. The residual oil of the larval fat biomass after each extraction was subjected to subsequent extraction under the same condition as the primary extract. This procedure possibly enhanced the hydrolysis and crystallization, which led to liberating a significant amount of USFAs observed in the AWME3. 

Sequential extraction (SE) is mainly based on the polarity and acidity used to remove the polar nonphenolic compounds such as sugars and organic acids. In the present study, the SE used the same extraction reagent, and there was a significant change in the content and the composition of FAs detected in the SEs. In addition, the sequential extraction introducing products with higher purity and -greater activity [55]. Mai et al. (2019) [56] determined the content of FAs in crude and refined larvae oil obtained from BSFL, noticing a significant change occurred in the content of FAs, where the percentages of FAs such as palmitoleic (16:1), oleic (18:1) increased more in refined than crude oil and that was matching with our study. 

The prominent fatty acids profile of *H. illucens* larvae fat SEs are presented in Table 4. In our new consecutive extraction, the most abundant USFAs was cis-oleic acid (C18:1 n-9), representing 22.65–26.28% of the total fatty acids, followed by 3.03–3.15% of palmitoleic acid (16:1), and 5.27–6.62% of SFA—myristic (C14:0) acid. These results are in line with 18.24%, 2.36%, and 6.56%, respectively, obtained by Rabani et al. (2019) [57], who used a hard-to-reproduce extraction method based on a proprietary biological decomposer and 5% acetic acid (to adjust pH to 6) to separate Hi larvae lipids without pressing or warming. In contrast, our extraction procedure significantly enriched SFA, the leading group of fatty acids in all SEs (Table 4), including the most abundant (21.76–26.19%) palmitic (C16:0), and 5.82–5.93% stearic (C18:0) acid vs. 16.27% and 1.43% [57], respectively. Although our procedure allowed less isolation of lauric acid (C12:0)- 17.66–19.32% vs. 40.79%. Our results also corroborated earlier findings obtained [40,45,58,59]. Ewald et al. (2020) [60] reported higher percentages (3–13%) of linoleic acid (C18:2) compared to <1% found in our SEs content. Even though AWME1 contains the highest percentages of SFAs, its antibacterial activity was the lowest among other extracts against the fish pathogenic bacteria strains.

During our SE, new potentially bioactive molecules were liberated and detected in AWME3: 9,12-hexadecadienoic acid, methyl ester-0.25%; cis-9-hexadecenal-0.13%; 2,4-dodecadienal, (E, E)-0.13%; lauric acid beta-monoglyceride-1.08%, and cis-oxiraneundecanoic acid, 3-pentyl-,methyl ester -1,14%. Albeit the yield of extract decreased gradually from 60 to 30 mg in AWME3, the amount of USFAs, particularly cis-oleic acid (18:1), increased from 22.65% in the first to 26.28% in the third SE (Table 4). These findings were in line with Imbimbo et al. [61], who extracted 2.5% and 34% of polyunsaturated fatty acids (PUSFAs) from the raw biomass of *Galdieria phlegrea* and the residual biomass, respectively. Of note, the glycerol percentages increased gradually from 0% (AWME1) to 7.87% (AWME3), while Ushakova et al. (2016) [62] detected traces (0.36%) of glycerol in *H. illucens* larvae oil.

It might be helpful to understand the chemical reactions that occur through the SE processes of the *H. illucens* larvae fat. The physical and chemical properties of lipids depend on the composition of their fatty acids, the length and saturation degree of the carbonic chain, and their melting points. Thermal hydrolysis takes place mainly within the oil phase rather than on the water-oil interface. Hydrolysis is more desirable in oil with short USFAs, such as cis-9-hexadecenoic acid, octadec-9-enoic acid (cis-oleic), than oil with long SFAs, like stearic, myristic, and lauric acid, because short USFAs are more soluble in water than long saturated fatty acids. The long-time contact between the oil and the aqueous phase of the larval fat increases hydrolysis of BSFL oil. Besides, a large amount of water (90%) in our extraction reagent might rapidly increase larval oil’s hydrolysis [63,64]. 

There are three Equations (1)–(3) controlled and represent the flow of chemical reactions which occur continuously during the conventional extraction method [65].
RCOOR′ + H_2_O ⇌ RCOOH + R′OH  (Hydrolysis)(1)
RCOOH + R′OH ⇌ RCOOR′ + H_2_O  (Esterification)(2)
RCOO R′ + R″OH ⇌ RCOO R″ + R′OH (Alcoholises)(3)

Following our SE processing, FFAs were liberated from BSFL fat during several steps, including hydrolysis, sonication, and homogenization under the condition of slow hydrochloric acid catalysis. Furthermore, the acid catalysis in water promoted the hydrolysis reaction in the sequential extraction. Besides, several treatments during the sequential extraction of the larval fat led to limited modifications observed in the composition of the third extract compared with the other SEs. USFAs presented in the various SEs were octadec-9-enoic acid (cis-oleic acid), 9,12-octadecadienoic acid (Z, Z)- (linoleic acid), and cis-9-hexadecenoic acid (palmitoleic acid) (Table 4). To release USFAs from the solid fat, we used methanol for esterification of larval fat to be dissolved in the aqueous phase at 52 °C followed by a complete extraction for 24 h at room temperature (20 °C). The temperature decrease led to the crystallization of the SFAs during the first round of extraction to produce various levels of FAs. A proportion of SFAs were also homogenized and interfaced with the aqueous phase based on their melting points through hydrolysis. The crystallization process was visible after storage of these concentrated SEs in the refrigerator at 4 °C, where SFAs were crystallized on the surface while the USFAs were soluble in the extraction solution [65]. These USFAs possessed antimicrobial activity against Gram-negative and Gram-positive bacteria [66,67,68]. Furthermore, they were effective against multidrug-resistant bacteria [62]. 9,12-hexadecadienoic acid, methyl ester, cis-9-hexadecenal, 2,4-dodecadienal, (E, E)-, dodecanoic acid, 2-hydroxy-1-(hydroxymethyl) ethyl ester (lauric, acid beta. -monoglyceride), oxiraneundecanoic acid, and cis- 3-pentyl-, and methyl ester, the new compounds GC-MS detected in AWME3, are known to possess antimicrobial activity against different pathogenic bacteria strains [69,70].

Our data suggested that fatty acids, the main constituents of oils and fats, might be partially esterified to glycerol during the sequential extraction. Indeed, AWME3 was enriched in glycerol by 7.87%, compared to AWME1 by 0% and AWME2 by 3.47% (Table 4). The high amount of glycerol possibly resulted from the high hydrolysis rate of the remaining oil, where the hydrolysis rate increased. The esterification by methanol using HCl as catalyst allows more access for methanol and the acid to hydrolyze the residual oil to produce glycerol as a by-product along with FFAs. In addition, the residual biomass of the larval fat was in the liquid state, promoting hydrolysis in the second and the third round of extraction [71]. Glycerol evaporates at 150 °C, while SEs are concentrated under a vacuum at 45 °C. Hydrolysis was carried out several times during the SE for the same biomass, which led to the accumulation of glycerol in the oil phase and promotion of the monounsaturated FFAs production by hydrolysis reported previously [72]. Di- and monoacylglycerols, glycerol, and FFAs presented in the oil phase accelerate the further hydrolysis reaction of oil [73]. The positional and geometric isomers predominately presented in FAs in the cis-isomers form, in natural lipids promote the crucial role of FAs in killing microbes. Oxygenated functional groups of naturally occurring fatty acids include the hydroxyl, keto, and epoxy groups. The hydroxy-substituted acids are most commonly abundant in FAs. Oleic acid is the most widespread and is an article of commerce and a component of all dietary fat while antibacterial effective against pathogenic bacteria [74]. Noteworthy, our improved extraction procedure enabled significant enrichment of this valuable component in AWME3 (Table 4) compared to other published methods of Hi larvae fat extractions. 

Livestock activities of fish farming are affected by infectious diseases, mainly associated with bacteria *Aeromonas* spp., which have been linked with mortality ranged between 50% and 100% in aquaculture production systems. Therefore, the antibacterial properties of obtained SEs were determined by several assays against pathogenic fish bacteria. Our data demonstrated that the activity of the SEs from the BSFL fat increased dramatically due to these consecutive extractions. AWME3 turned out to be the most active among the different SEs obtained by exploring the same biomass of BSFL fat. Furthermore, *A. hydrophila* and *A. salmonicida* were inhibited and killed at a low dose of AWME3. The activity of the three mixed SEs was less than AWME3 alone. We suspect that AWME3 possessed the highest antibacterial activity against tested strains because of the high amount of oleic acid (26.28%), but not due to glycerol (7.87%), albeit the antimicrobial activity of the last one was reported [75]. We tested the same proportion of pure glycerol (7.87%) by mixing with the AWME1 under the aseptic condition to give the final concentrations 40 mg/mL of the glycerol-AWME1 mixture. The resulted IZD value of glycerol-AWME1 mixture was not significantly different from AWME1 alone against pathogenic fish bacteria (data not shown). This result corroborates the study of Saegeman et al. [76] that reported only higher concentrations of glycerol (85%) and higher temperatures of incubation had a long-term antimicrobial effect and virucidal activity on several types of viruses. On the other hand, glycerol molecules react with fatty acids like lauric acid and form glycerol monolaurate (GML). In our study, AWME3 contains 32.12% compounds with one or more cis-double bonds, which proved to increase the antibacterial activity of fatty acids and their derivatives, as reported by Yoon et al. [77]. 

In the present study, an antibiotic susceptibility assay was performed against *A. hydrophila* and *A. salmonicida*. Both tested *A. hydrophila* and *A. salmonicida* strains demonstrated multidrug-resistant capacity (MDR). *A. salmonicida* was more resistant than *A. hydrophila* against six different groups of antibiotics, including penicillin, vancomycin, erythromycin, doxycycline, rifampicin, and colistin. *A. hydrophila* was moderately sensitive to colistin (CT) and resistant to doxycycline (DOX). Of note, both strains were 100% resistant to penicillin and vancomycin without visible IZD. The intermediate sensitivity to penicillin-streptomycin (P/S) was observed for both strains. Thus, *A. salmonicida* was resistant to 60% while *A. hydrophila* was resistant 50% of the total ten antibiotics used in this test. Interestingly, treatment with rifampicin, erythromycin, and chloramphenicol showed the sharpest decreasing IZD after 24 h of incubation at *A. hydrophila* and *A. salmonicida* strains (Table 1, Appendix A). In this regard, our data significantly complement and corroborate with a previous study [78] of the antibiotic sensitivity of the same bacteria genus (*Aeromonas* spp.) that also observed resistance to ampicillin.

Here, for the first time, we demonstrated that AWME3 isolated from BSFL fat inhibited and killed bacteria at the lower dose MIC (0.095–0.38 mg/mL) and MBC (0.19–0.38 mg/mL), compared to the active compounds extracted from *Salix babylonica* that inhibited the growth of *A. hydrophila* at MIC = 25 mg/mL [79]. AWME3 was also more efficient than the methanolic extract from seeds and flowers of *Michelia champaca* that inhibited *A. hydrophila* growth at 31.3 and 15.6 mg/mL, respectively [80]. Our AWME3 extract was more active and potent than marine extracts of *Diadema setosum* (sea urchin) that contain polyunsaturated fatty acids, and recorded IZD 18.33 ± 0.58 mm at a concentration 50 mg/mL, while the MIC and MBC were 6.25 and 12.5 mg/mL, respectively against *A. hydrophila* [81]. The MIC for cinnamaldehyde was in the range 128–512 µg/mL, meanwhile MBC values ranged between 256 to 1024 µg/mL against different isolates of *A. hydrophila* [82]. Emodin could eradicate the *A. hydrophila* at 2.5 mg/mL [83], while oleic acid inhibits both Gram-negative and -positive bacteria at a concentration of 0.5 mg/mL after 15 min treatment [77]. Notwithstanding, antibacterial substances from the metabolites of *Vitis rotundifolia* (Muscadine) roots [84] demonstrated MIC between 10 and more than 100 µg/mL against pathogenic fish bacteria and MIC_50_ values fluctuated between 16.5 and 22 µg/mL, albeit they were only bacteriostatic. In the present study, AWME3 was the most effective as an antibacterial substance from BSFL fat and could eradicate and inhibit the pathogenic fish bacteria at a low dose. The high AWME3 activity could embark an alternative for antibiotics in aquaculture habitats; consequently, it has a potential to eradicate and treat the MDR fish bacteria.

The suggested antimicrobial FFAs mechanism of action could be related to their amphipathic nature with non-cylindrical molecular geometry. Their ability to form micelles in solutions and incorporate them readily into lipid membranes will increase the curvature stress within the lipid bilayer [85]. Consequently, FFAs can efficiently create instabilities in the lipid membrane and diminish the membrane permeability barrier. The disconcerting membrane effect of fatty acids is a complex and concentration-dependent process, which is strongly modulated by the chemical structure of the fatty acid, e.g., chain length and degree of unsaturation, such as the phase state of the lipid bilayer [86]. The effect of unsaturated fatty acids (e.g., oleic acid) on the permeability barrier of liposomes was, in general, more active than saturated fatty acids (e.g., palmitic acid, C16) [85]. While being out of the scope of the current study, the detailed mechanism of the observed antibacterial AWM3 effect awaits our further ongoing studies.

For many years, antibiotics have been used indiscriminately in aquaculture, which has led to a high rate of resistant or multi-drug-resistant bacteria. This problem has boosted the search for functional alternatives with a safe environmental impact. However, it is reasonable to raise concerns about the high risk of fish exposure to pharmaceuticals, as it may threaten human health through the diet. Release of the antibiotics into the aquatic environments causes a chronic risk to fish and can cause genetic and histologic alterations in fish tissue. In addition, antibiotics in fish tissue that humans consume may potentially pose a threat to human health. Furthermore, antibiotics rank higher than other compounds detected in aquatic environments, such as sex hormones, cardiovascular, and antineoplastic agents. Besides the mortality, genotoxic properties consider as indicators of aquatic toxicity in fish due to the overuse of antibiotics. Antibiotics are ultimately presumed to inhibit the survival, development, and hatching rates of fish, by predominantly disrupting the intracellular redox balance and inducing oxidative stress [87]. 

Tetracycline and its derivatives, such as doxycycline and oxytetracycline, are the most threatening antibiotics group in the ecosystem. They are commonly used in aquaculture to treat bacterial fish diseases such as vibriosis, flavobacteriosis, and erythrodermatitis. Doxycycline inhibits protein synthesis by disrupting the association of aminoacyl-tRNA in bacterial ribosomes. Moreover, this antibiotic is hardly removed from wastewater treatment plants and is known for its non-target effects in aquatic environments. In addition, it is reported that doxycycline exerts genotoxic effects in rainbow trout through the induction of DNA damage and chromosome breaks during acute and chronic exposure. At low concentrations 20–200 mg/L it affects hematological markers such as hemoglobin, hematocrit, and red blood cell count, as well as the activity of enzymes such as aspartate aminotransferase (AST), alanine aminotransferase (ALT), and dehydrogenase (LDH), in carp. Moreover, it causes DNA damage in erythrocytes when exposed to juvenile tilapia at an environmentally relevant concentration of 4 μg/L. It is proved that doxycycline or oxytetracycline can delay hatching when exposed to zebrafish embryos with an EC_50_ of 127.6 mg/L. Doxycycline also caused intestinal damage with inflammatory responses, and prolonged exposure (0.1–10,000 μg/L) caused alteration in swimming pattern and affected the intestinal bacterial community along with oxidative problems in zebrafish. Doxycycline proved to causes changes in the intestinal morphology and microbiota in tilapia species [87]. 

BSFL contains a high amount of protein and fat content, anti-pathogenic and anti-inflammatory properties connected with antimicrobial peptides, lauric, palmitic, myristic, stearic, arachidic, and capric acid that were stated to be antimicrobial compounds [52,88,89,90]. Lipids and their derivatives comply with antimicrobial characteristics and have been shown to induce antibacterial activity during in vitro and in vivo studies, potentially leading to finding new natural alternatives beneficial in fisheries units [91]. Once antibacterial mechanisms of AWME3 active ingredients mode of action will be identified in our future studies, their antibacterial activity will be tested separately or in combinations. Subsequently, assessing their cytotoxicity on mammalian cell lines will warrant the possible safe use of these compounds in humans. 

## 4. Materials and Methods

### 4.1. Chemical Reagents, Bacterial Strains and Supply

Fat separated from alive *H. illucens* larvae of 15 days old compressed by mechanical pressing machine, provided by “NordTechSad, LLC” company (Arkhangelsk, Russia) and used for this study. Reagents for extraction solution, including hydrochloric acid (HCl), methanol (CH_3_OH) were purchased from Thermo Fisher Scientific, Waltham, MA, USA, and Milli-Q water was obtained from Water System, Ultrapure, Millipore, Direct-Q^®^ 3 with UV. Luria-Bertani (LB) broth and agar, Mullar Hinton (MH) agar were purchased from Sigma-Aldrich, St. Louis, MO, USA. Tissue culture 96-well plates (TPP, Trasadingen, Switzerland), Petri dishes (90 mm) (Pertin, Saint Petersburg, Russia), paper disks with size 6.0 mm diameter (Himedia, Mumbai, India), sterile swab (Nigbo Greetmed medical instruments CO., LTD., Nigbo, China) were used for this work. Antimicrobial susceptibility disks (6 mm) with 100 U/mL-100 µg/mL/disk penicillin-streptomycin (P/S), 100 µg/mL/disk Doxycycline (DOX) (Zoofarmagro SRL, Chişinău, str. Cameniţa, Moldova), 1000 µg/mL/disk Chloramphenicol (Ch), 200 µg/mL/disk Gentamycin (G) and 25 µg/mL/disk Amphotericin B (Am)) were purchased from Gibco, Thermo Fisher Scientific, Waltham, MA, USA. Discs with 2U/disk Penicillin (P), 5 µg/disk Vancomycin (VA), 60 µg/disk Erythromycin (E), 15 µg/disk Rifampicin (RD), 1000 µg/disk Kanamycin (K), and 10 µg/disk Colistin (CT) were purchased from Oxoid, Basingstoke, Hampshire, United Kingdom. 

Bacterial strains used in this study were *Aeromonas hydrophila* (ATCC 49140), and *Aeromonas salmonicida* (ATCC 33658) were purchased from the American Type Culture Collection (ATCC) Manassas, USA. The bacterial strains were stored in glycerol stock (30%, *v/v*) at −80 °C. To culture, they were incubated overnight in 10.0 mL of LB broth at 26 °C and shaking at 200 rpm/min. The overnight culture was adjusted to 0.5 McFarland standard (1 × 10^8^ CFU/mL) and used in susceptibility tests. All experiments were performed under antiseptic conditions (Safe Fast Elite, Ferrara, Italy).

### 4.2. BSFL Fat Sequential Extraction

SEs sequential extractions were performed as previously described [47] with minor changes. Briefly, three grams of compressed *H. illucense* larvae were melted in extraction solution (H_2_0: CH_3_OH: HCl) with ratio 90:9:1% *v/v*, respectively at 52 °C under hot tap water for five minutes, then subjected to vigorous vortexing. The homogenized emulsion was subjected to continuous extraction for 24 h using the orbital shaker Mixmate (Eppendorf AG, Hamburg, Germany) at room temperature. After that, the mixture was sonicated at 35 °C for 10.0 min (Elmasonic S 30H, Singen, Germany) and finally vigorously homogenized using ULTRA TURRAX-25 homogenizer (IKA, Staufen, Germany) for 10 min. The final homogenate was centrifuged at 4000× *g* (Centrifuge 5804, Eppendorf AG, Hamburg, Germany) for 20 min at room temperature to separate the insoluble fat.

After the first round of extraction, the collected supernatant was named AWME1. The remaining oil layer was subjected to the second round of extraction using the same protocol. The collected supernatant after the second extraction was named AWME2. The remaining oil phase was treated third time in the same way, and the collected supernatant was nominated as AWME3. Finally, all extracts were concentrated under the vacuum (Concentrator plus, Eppendorf AG, Hamburg, Germany) at 45 °C for 13 h and then stored at 4 °C until use.

### 4.3. Antibiotic Susceptibility

Antibiotic sensitivity determined using the disk diffusion assay in Muller–Hinton agar. Following Kirby–Bauer method [92]. The pure cultures of *A. hydrophila* and *A. salmonicida* were incubated overnight at 26 °C and shaking at 200 rpm, then bacterial density adjusted to 0.5 McFarland standard (1 × 10^8^ CFU/mL). The bacterial suspension streaked in three planes with a cotton swab on the surface of MH agar plates. Sterile disks with 6 mm diameters were impregnated with 50 µL of penicillin-streptomycin (100 U/mL-100 µg/mL/disk), Gentamycin (200 µg/mL/disk); Doxycycline (100 µg/mL/disk); Chloramphenicol (1000 µg/mL/disk), then these disks dried in ambient conditions for 30 min. Penicillin-streptomycin and gentamycin were prepared and diluted in distilled water, while chloramphenicol was prepared and diluted in absolute ethanol under sterile conditions. The inhibition zone diameters (IZD) measured, compared to the measures of the CLSI guidelines [93]. The bacteria were divided into three groups of sensitive, intermediate, resistant to antibiotics, then resistant to three or more antimicrobial agents were classified to be multidrug-resistant strains. All data values recorded at 12 h and 24 h, respectively. 

In addition, the multiple drug resistance (MDR) index was calculated for each strain. The MDR index was calculated as “a/b”, where “a” is the number of antibiotics to which a particular tested bacterium is resistant and “b” is the total number of antibiotics tested for this strain [94].

### 4.4. Disk Diffusion Assay

This assay was performed separately for each SE (AWME1, AWME2, and AWME3), and a mixture of them was prepared at equal volumes with 40 mg/mL of concentration, using the previously described method [95]. Mueller–Hinton agar plates were seeded with inocula grown to reach the log phase of the bacterial strains in LB broth at 26 °C under shaking. The inoculum suspension was diluted and adjusted to 1 × 10^8^ (CFU/mL). Then, 50 µL of each SE of larvae fat was applied to sterile paper 6 mm disks to give the final concentration of 40 mg/mL. All disks dried under ambient and sterile conditions, then placed on the surface of the inoculated plates. The plates incubated at 26 °C for 24 h. Antibacterial activity was determined by measuring the diameter of the inhibition zone formed around the disk. Likewise, the antibacterial activity for the most effective SE was determined at 1.88, 3.75, 7.5, 15, 30, and 40 mg/mL concentrations. The IZD values recorded at 12 h and 24 h of incubation time. The 50 µL of the negative control (extraction reagent) was also used to determine its effect on the tested bacteria. The 50 µL of DOX with concentration 100 µg/mL used as a positive control. All the samples were done in duplicates at three independent experiments.

### 4.5. MIC and MBC Determination for BSFL Fat Extracts

The Minimum Inhibitory Concentration (MIC) was determined by microdilution assay methods in accordance with [50]. The 100 µL of fresh LB broth were transferred gently to a 96-well plate (TPP, Trasadingen, Switzerland). Then, 100 µL of each SE of BSFL fat was separately introduced to the LB broth wells and diluted by two-fold serial dilutions, then 100 µL of bacterial culture with density 10^6^ CFU/mL transferred to all the wells of the microplate except the blank (with LB broth only). The final concentrations of AWME1 were 0.046, 0.092, 0.187, 0.375, 0.75, 1.5, 3, and 6 mg/mL, AEME2 were 0.058, 0.117, 0.234, 0.468, 0.937, 1.87, 3.75, and 7.5 mg/mL, AWME3 were 0.012, 0.024, 0.048, 0.095, 0.19, 0.38, 0.75, and 1.5 mg/mL, and mixtures of these SEs were 0.0458, 0.092, 0.183, 0.367, 0.73, 146, 2.93, and 5.87 mg/mL. The bacterial suspension was adjusted to 5 × 10^5^ CFU/mL by measuring the OD_600_ using a CLARIOstar microplate reader (BMG LABTECH, Ortenberg, Germany). Likewise, the positive control (DOX) was tested against fish pathogenic bacteria with final concentrations ranged between 0.195 and 25 µg/mL against *A. salmonicida*, while it was ranged from 0.008 µg/mL to 1.0 µg/mL against *A. hydrophila*. All micro-plates sealed and incubated at 26 °C for 24 h and shaking at 210 rpm. The minimum concentration of the larvae fat extracts and the antimicrobial agent that inhibited the growth of the tested strains was identified as the MIC. All experiments were performed in triplicate and measured at three different times to test reproducibility.

The minimal bactericidal concentration (MBC) defined as the minimum concentration, which kills 99.9% of the starting inoculum. The MBC was determined according to CLSI guidelines and Kot et al. [50,96]. The aliquot of 40 µL from the non-bacterial growth wells were transferred to MH agar plates and incubated at 26 °C for 48 h. The MBC was defined as the lowest concentration of the tested SE of BSFL fat that showed no growth on the surface of the Petri dishes agar. The MBC of each SEs compared with the MBC of the positive control (DOX) used against fish pathogenic bacteria. All tests were performed in triplicates, and each experiment repeated independently three times.

### 4.6. Growth Curve Analysis and the Half of the Minimum Inhibition Concentration (MIC_50_)

The growth curve of tested bacteria strains was plotted based on the turbidimetric assay of the most effective AWME3 of BSFL fat and the antibacterial agent (DOX) according to Choi and Jiang [54] with slight modification. The 100 µL of AWME3 was added in the 96-well plate containing 100 µL of sterile LB broth and then was subjected to the 2-fold serial dilutions method. Finally, 100 µL of tested strains with bacterial density 10^6^ CFU/mL was introduced to the microplate up to the nine well except the sterile blank (LB broth only). The final concentration of AWME3 was in the range of 0.012–1.5 mg/mL, and the final density for bacterial strains was 5 × 10^5^ CFU/mL. Likewise, the positive control (DOX) was subjected to the same procedure to obtain the final concentration of 0.008–1.0 µg/mL for *A. hydrophila* and from 0.19 µg/mL to 25 µg/mL against *A. salmonicida*. Then, all microplates were sealed and incubated at 26 °C for 24 h and shaking at 210 rpm. The OD_600_ values recorded intervals every 2 h during 24 h, all tests performed in triplicate, and each experiment repeated three times. 

The half of the minimum inhibition concentration (MIC_50_) determined at 6 h, 12 h, and 24 h by measuring the OD_600_ of tested strains using CLARIOstar microplate reader (BMG LABTECH, Ortenberg, Germany). In addition, the IC_50_ values of the positive control (DOX) were evaluated in parallel by the same conditions against the fish pathogens. Graph pad prism version 7.0 software used to calculate the IC_50_ by non-linear regression. All experiments conducted in triplicate and repeated three times. 

### 4.7. Gas Chromatography-Mass Spectroscopy Analysis (GC-MS)

The SEs from BSFL fat analysed separately by GC-MS-QP2010 ultra mass spectrometer (Shimadzu, Canby, CA, USA). The instrument composes of an autosampler and gas chromatography interfaced to a mass spectrometer. The capillary column DB-5 ms measuring (30 m × 0.25 mm) with a thickness of 0.25 mm coated with non-polar silphenylene polymer with the polarity of 5% diphenyl and 95% dimethylpolysiloxane stationary phase (Restek, Bellefonte, PA, USA). The helium used as a carrier gas with a linear flow rate from 1.0 to 15 mL/min, and the column head pressure was 50.4 kPa. The operating procedure was as follows: 1 µL of every SE was injected by the autosampler injector automatically. The injector and detector temperatures maintained at 280 °C and 250 °C, respectively. The temperature program was initially set at 40 °C, held for 1 min, and then it was increased to 210 °C at a rate 15 °C/min, held for 0 min, then it was increased to 216 °C with a rate 5 °C/min and held for 0 min. Then, the temperature was increased to 300 °C with a flow rate of 40 °C/min and held for 14.87 min. The GC–MS values were analyzed using electron impact ionization at 70 Ev. The content of each SE was identified based on a comparison of their retention time (min), peak area, peak height, and mass-spectral patterns with those spectral databases of authentic compounds stored in the National Institute of Standards and Technology (NIST) library. Compounds with chromatogram peaks matched with similarity index (SI) ≥70% in NIST-8 library were ascertained. 

### 4.8. Statistical Analysis

Statistical analysis was performed using Repeated Measurements of two-way variance analysis (Two-way RM-ANOVA) to determine the significant statistical differences between treatments. The difference between means was compared with Tukey’s test (*p* < 0.05), and all data assessed by the standard deviation (SD) and standard error of the mean (SEM) using the statistical software Graph Pad Prism version 7.0 (San Diego, CA, USA).

## 5. Conclusions

In the present study, we improved our extraction procedure by using three consecutive extractions of the same biomass of BSFL fat using the acidic water–methanol solution. We demonstrated that the major extracted constituents were free fatty acids (FFAs) and fatty acids derivatives, and the sequential extraction continuously improved their antibacterial activity. For the first time, the antimicrobial susceptibility testing demonstrated the high dose-dependent antibacterial activity of each SEs against the most important pathogenic fish bacteria *A. hydrophila* and *A. salmonicida*, especially antibiotic-resistant *A. salmonicida*. The bacteriostatic and bactericidal (MIC/MBC) activity of SEs was significantly enhanced through the sequential extraction of the same BSFL fat sample. The third sequential extract AWME3 possessed the strongest antimicrobial activity compared to AWME1 and AWME2. Besides, via our improved procedure, we were able to enrich gradually the unsaturated fatty acids (USFAs) content in our SEs. Thus, the larvae fat from *Hermitia illucens* may serve as an excellent reservoir of bioactive molecules with a good capacity to eradicate the MDR bacteria, thus having promising potential for practical application against *Aeromonas* spp. in the aquaculture field, as an alternative for antibiotics.

## Figures and Tables

**Figure 1 ijms-22-08829-f001:**
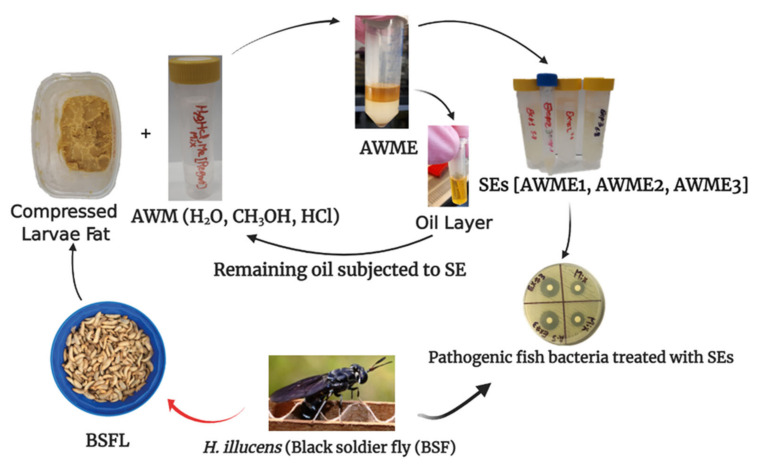
SEs stages of antibacterial compound isolation procedure from *H. illucens* larvae fat.

**Figure 2 ijms-22-08829-f002:**
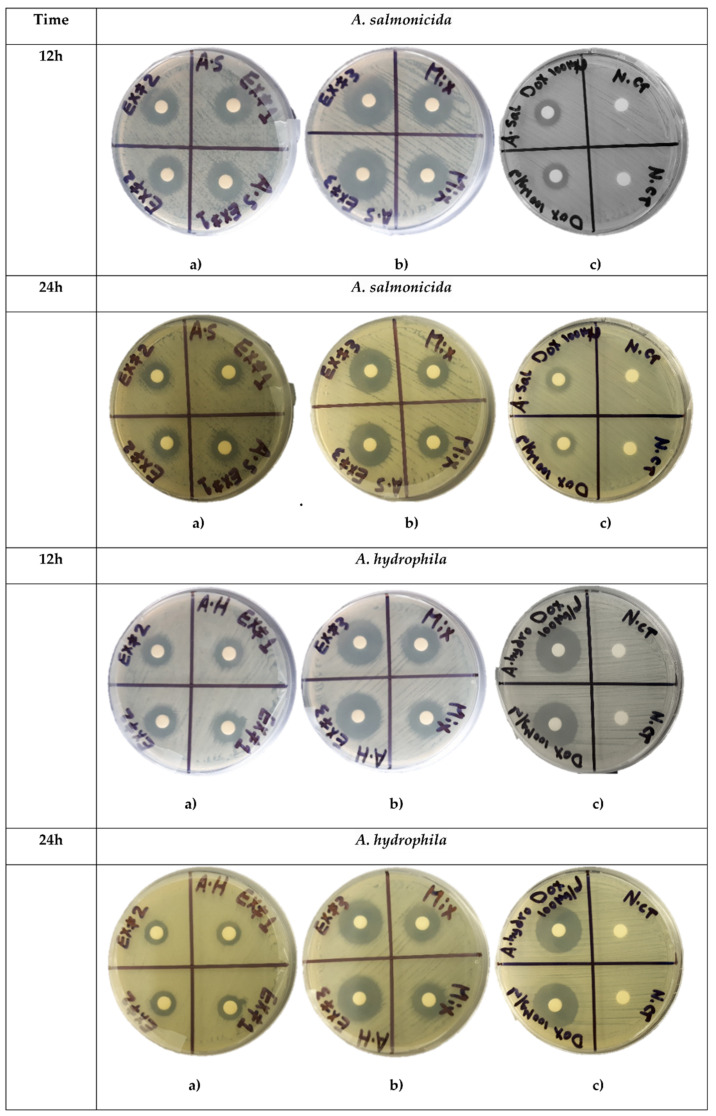
Susceptibility of *A. hydrophila* and *A. salmonicida* to three sequential extracts treatment determined by disk diffusion assay. Disks loaded in duplicates with 50 µL of (**a**) AWME1 (Ex#1) and AWME2 (Ex.#2); (**b**) AWME3 (Ex.#3) and a mixture of the three SEs (Mix) at concentration 40 mg/mL; (**c**) Positive control (Doxycycline, 100 µg/mL) and the extraction reagent as a negative control (N.ct).

**Figure 3 ijms-22-08829-f003:**
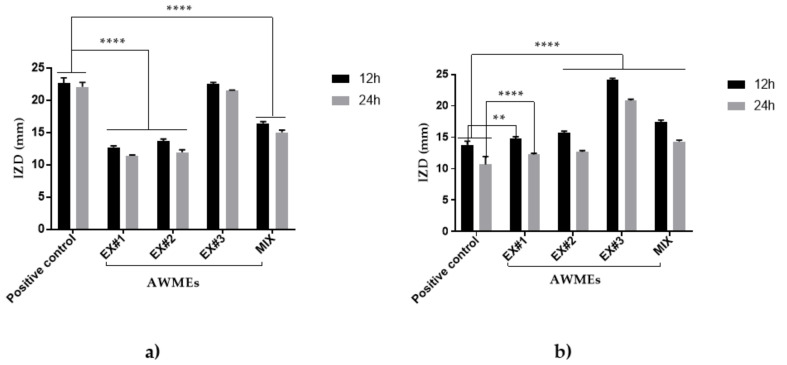
Susceptibility of (**a**) *A. hydrophila* and (**b**) *A. salmonicida* bacteria strains to various SEs measured by Disk diffusion assay. The bacteria subjected to AWME1 (Ex#1), AWME2 (Ex#2), and AWME3 (Ex#3), respectively, and a mixture of three extract (Mix) adjusted to the same 40 mg/mL concentration. The IZD values calculated after 12 h and 24 h of incubation at 26 °C by measuring the diameter of inhibition zones (IZD) surrounding of the discs (in mm). Doxycycline (DOX) 100 µg/mL used as a recommended antibacterial positive control. All values are represented as mean ± SD, in triplicate (*n* = 3). All results analyzed by two-way ANOVA, followed by Tukey Multiple Comparisons Test. Data represented as a significant difference when it was compared with the positive control, and *p*-Value was ranged between ** *p* = 0.0047 and **** *p* < 0.0001.

**Figure 4 ijms-22-08829-f004:**
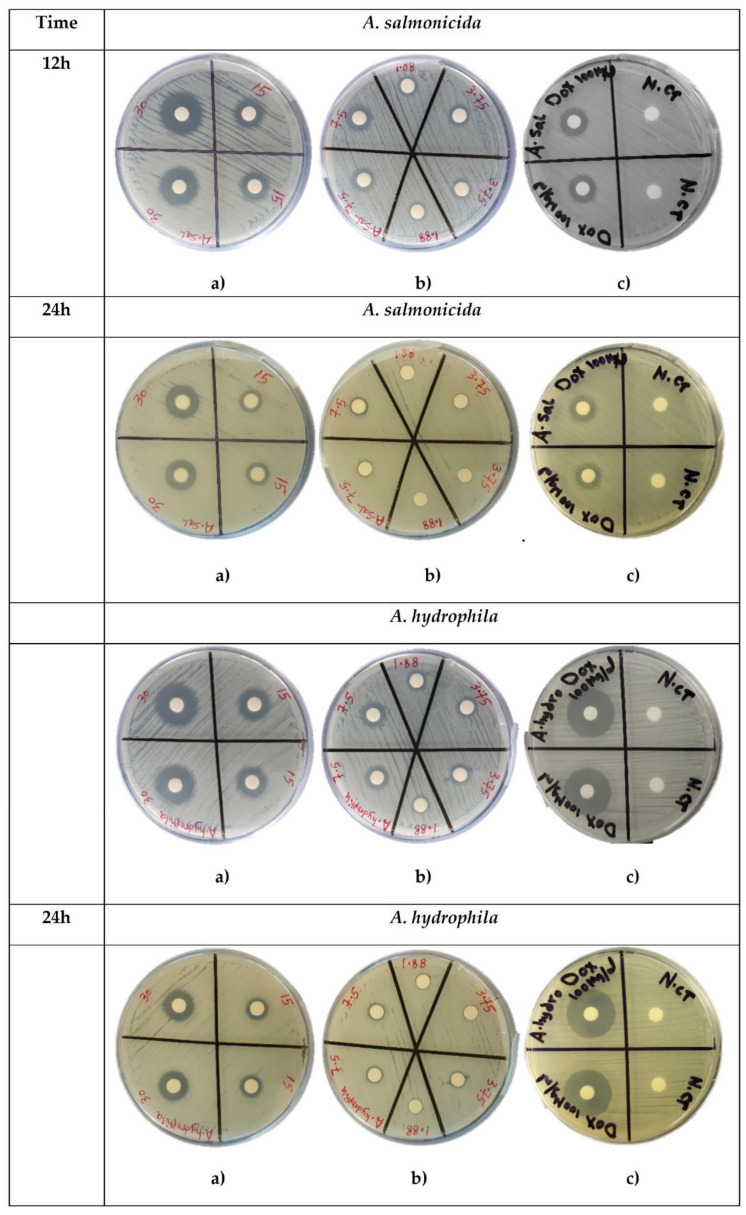
Dose-dependent susceptibility of *A. hydrophila* and *A. salmonicida* determined by disk diffusion assay of the AWME3. (**a**,**b**) IZD formed around the disks loaded with AWME3 at tested concentrations in the range of 1.88–40 mg/mL after 12 h and 24 h of incubation; (**c**) Positive control (Doxycycline 100 µg/mL) and negative control (extraction solution of AWM mixture, N.ct) determined at 12 h and 24 h of incubation. Mean IZD for each condition recorded in three independent experiments.

**Figure 5 ijms-22-08829-f005:**
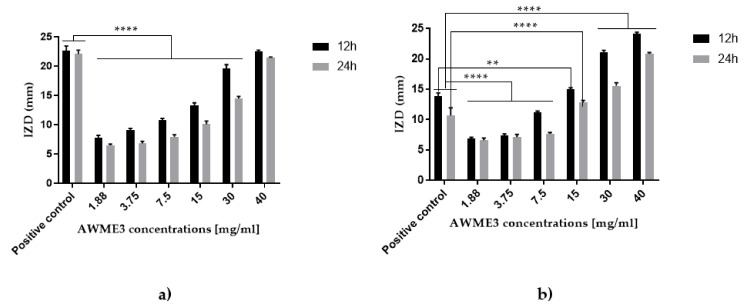
Susceptibility of *A. hydrophila* (**a**) and *A. salmonicida* (**b**) determined by disk diffusion assay of AWME3 (the third sequential extract) of BSFL fat. The bacteria strains were subjected to concentrations of 1.88, 3.75, 7.5, 15, 30, and 40 mg/mL of AWME3 from BSFL fat. The IZD were measured after 12 h and 24 h of incubation at 26 °C by measuring IZD surrounded of the disks. Doxycycline (DOX) used as the recommended positive antibacterial control. All values are represented as mean ± SD of three (*n* = 3) independent experiments. Data were analyzed by two-way ANOVA, followed by Tukey Multiple Comparisons Test. Data represented as significant difference as compared to positive control, and *p*-Value was ranged between ** *p* = 0.0019 and **** *p* < 0.0001.

**Figure 6 ijms-22-08829-f006:**
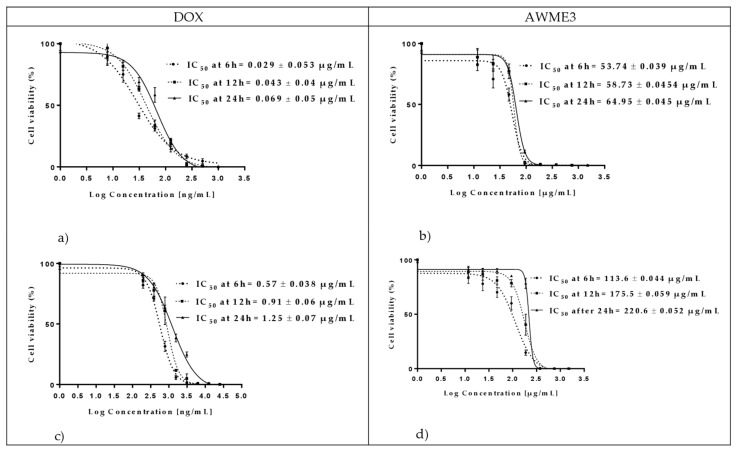
The MIC_50_ of fish pathogenic bacteria treated with AWME3 from the larvae fat in comparison with the Doxicycline (control antibiotic). The MIC50 was determined based on the turbidimetric assay data and compared to the positive control doxycycline (DOX). The pathogenic cells turbidity assessed at 6 h, 12 h, and 24 h of incubation. *A. hydrophila* treated with (**a**) doxycycline (DOX) and (**b**) AWME3; *A. salmonicida* treated with (**c**) doxycycline (DOX) and (**d**) AWME3. MIC_50_ values calculated by using a non-linear regression mode of Graph pad Prism7 software. Presented values are the mean ± standard error of the mean. The IC_50_ (MIC_50_) values as an average of three independent experiments ± standard deviation error mean (SEM) presented in Table 3.

**Figure 7 ijms-22-08829-f007:**
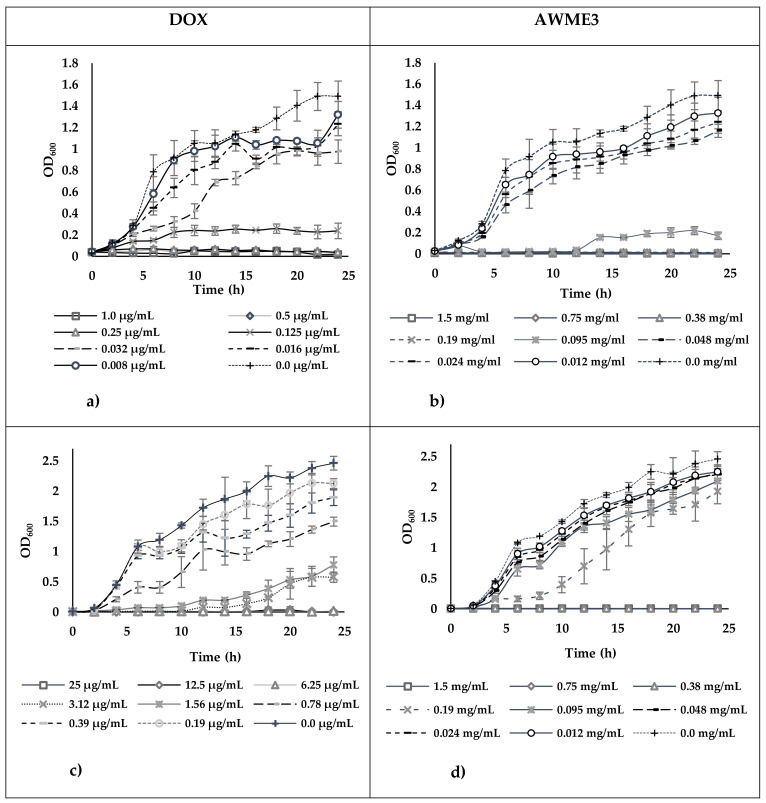
The growth curves of (**a**,**b**) *A. hydrophila* and (**c**,**d**) *A. salmonicida* treated with different concentrations of either AWME3, or DOX. Each data point is the average of three independent assays ± Standard deviation of the mean (SD).

**Table 1 ijms-22-08829-t001:** Susceptibility (IZD) of *Aeromonas* spp. to antibiotics.

	IZD (mm)						
Antibiotic	*A. hydrophila*		*A. salmonicida*		Breakpoint (mm)		
	12 h	24 h	12 h	24 h	R	I	S
G	16.4 (S)	16.2 (S)	17.33 (S)	17.1 (S)	≤12	13–14	≥15
Ch	22.68 (S)	22.18 (S)	25.15 (S)	22.58 (S)	≤12	13–17	≥18
K	24.65 (S)	24.08 (S)	27.48 (S)	26.1(S)	≤13	14–17	≥18
DOX	22.7 (S)	22.1 (S)	13.8 (R)	10.7 (R)	<17	17–18	≥19
P/S	12.75 (I)	12.2 (I)	13.88 (I)	13.87 (I)	≤11	12–14	≥15
CT	10.53 (I)	10.4 (I)	9.23 (R)	8.78 (R)	≤10	-	≥11
RD	14.08 (R)	11.85 (R)	14.93 (R)	12.93 (R)	≤16	17–19	≥20
E	12.63 (R)	10.33 (R)	12.78 (R)	12.53 (R)	≤13	14–22	≥23
VA	0 (R)	0 (R)	0 (R)	0 (R)	-	-	≥15
P	0 (R)	0 (R)	0 (R)	0 (R)	≤28	-	≥29

Abbreviations: G, gentamicin; Ch, chloramphenicol; K, kanamycin; DOX, doxycycline; P/S, penicillin-streptomycin; CT, colistin; RD, rifampicin; E, erythromycin; VA, vancomycin; P, penicillin; R, resistant; I, intermediate; S, susceptible; (0), means no inhibition zone around the disc on the plate; (-), not determined.

**Table 2 ijms-22-08829-t002:** MIC and MBC of SEs against fish pathogenic bacteria *Aeromonas* spp.

Antibacterial Extract		Concentration (μg/mL)	
	Parameter	*A. hydrophila*	*A. salmonicida*
AWME1	MIC	1500	1880
	MBC	1500	1880
AWME2	MIC	940	940
	MBC	1880	1880
AWME3	MIC	95	380
	MBC	190	380
MIX	MIC	730	940
	MBC	1460	940
Positive control (Doxycycline)	MIC	0.25	6.25
	MBC	0.5	12.5

Notes. MIC: Minimal inhibitory concentration; MBC: Minimal bactericidal concentration; MIX: Mixture of SEs in equal volumes of AWME1, AWM2, and AWME3 adjusted to the same concentration before treatment of *A. hydrophila* and *A. salmonicida* bacteria.

**Table 3 ijms-22-08829-t003:** The antimicrobial efficacy of AWME3 and doxycycline (DOX) against *Aeromonas* spp. at different times of incubation.

	MIC_50_					
Bacteria Species	DOX (µg/mL)			AWME3 (µg/mL)		
	6 h	12 h	24 h	6 h	12 h	24 h
***A. hydrophila***	0.029 ± 0.05	0.043 ± 0.0.04	0.069 ± 0.05	53.74 ± 0.039	58.73 ± 0.045	64.95 ± 0.045
***A. salmonicida***	0.57 ± 0.038	0.91 ± 0.06	1.25 ± 0.07	113.6 ± 0.044	175.5 ± 0.059	220.6 ± 0.052

**Table 4 ijms-22-08829-t004:** Major compounds of various SEs of BSFL fat detected by GC-MS.

Common Name		Chemical Structure	C:D	AWME1		AWME2		AWME3	
				Peak Area %	SI %	Peak Area %	SI %	Peak Area %	SI %
SFAs	Palmitic acid	CH_3_(CH_2_)_14_COOH	16:0	26.19	96	22.01	96	21.76	96
	Lauric acid	CH_3_(CH_2_)_10_COOH	12:0	19.32	97	16.68	97	17.66	97
	Myristic acid	CH_3_(CH_2_)_12_COOH	14:0	6.62	97	5.56	97	5.27	97
	Stearic acid	CH_3_(CH_2_)_16_COOH	18:0	5.93	93	5.74	94	5.82	94
	Arachidic acid	CH_3_(CH_2_)_18_COOH	20:0	0.34	88	0.46	91	0.31	90
	Capric acid	CH_3_(CH_2_)_8_COOH	10:0	0.31	89	0.26	86	0.3	84
	Pentadecyclic acid	CH_3_(CH_2_)_13_COOH	15:0	0.3	86	0.2	94	0.2	91
	Tridecyclic acid	CH_3_(CH_2_)_11_COOH	13:0	0.19	83	0.18	87	ND	ND
USFAs	cis-Oleic acid	CH_3_(CH_2_)_7_CH=CH(CH_2_)_7_COOH	18:1	22.65	95	23.9	95	26.28	95
	Palmitoleic acid	CH_3_(CH_2_)_5_CH=CH(CH_2_)_7_COOH	16:1	3.03	96	3.05	96	3.15	96
	Linoleic acid	CH_3_(CH_2_)_4_CH=CHCH_2_CH=CH(CH_2_)_7_COOH	18:2	0.37	91	0.21	89	0.21	91
	9-Octadecadienoic acid (Z)	CH_3_(CH_2_)_6_CH=CH(CH_2_)_7_COOH	17:1	ND	ND	0.26	88	ND	ND
FAs Derevatives	Glycerol	C_3_H_8_O_3_	-	ND	ND	3.47	95	7.87	96
	9,12-Hexadecadienoic acid, methyl ester	CH_3_(CH_2_)_3_CH=CHCH_2_CH=CH(CH_2_)_7_COO-CH_3_	-	ND	ND	ND	ND	0.25	78
	cis-9-Hexadecenal	CH_3_(CH_2_)_5_CH=CH(CH_2_)_7_CHO	-	ND	ND	ND	ND	0.13	83
	2,4-Dodecadienal, (E,E)-	CH_3_(CH_2_)_4_CH=CHCH=CH CHO	-	ND	ND	ND	ND	0.13	80
	Lauric acid beta-monoglycerol	C_15_H_30_O_4_	-	ND	ND	ND	ND	1.08	82
	Oxiraneundecanoic acid, 3-pentyl-, methyl ester, cis-	C_19_H_36_O_3_	-	ND	ND	ND	ND	1.14	85

Notes. SFAs: saturated fatty acids; USFAs: unsaturated fatty acids; ND: not detected; SI%: Similarity percentage for detected compounds based on NIST-8 library.

## Data Availability

Data available on request. The data presented in this study are available on request from the corresponding author.

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
