# Peer review of "Fatty Acids-Enriched Fractions of Hermetia illucens (Black Soldier Fly) Larvae Fat Can Combat MDR Pathogenic Fish Bacteria Aeromonas spp."

_ijms, 2021, doi:10.3390/ijms22168829_

Round 1
Reviewer 1 Report
The authors have studied the lipid content (in lipid fraction of larvae) of black soldier flies and its effect on pathogenic fish bacteria. It is a good study with a potential application in management of antibiotic utilization in aquaculture.
Major comments:
Discussion is to be developed and represented. I did not find a real discussion on the results. For example, it would be interesting if the authors can explain, even hypothetically, why the sequential lipid extraction can change the lipid content in unsaturated fatty acids.
It would also be interesting if authors compare their list of fatty acids with those of previous studies such as :
https://www.ncbi.nlm.nih.gov/pmc/articles/PMC6592434/
Minor comments
It would be more helpful if you briefly describe your lipid extraction method in M&M.
Table 4 , It would be easier to understand if you list fatty acids in D:C form beside their common names. for example :
Common name Chemical structure C:D
Palmitic acid CH3(CH2)14COOH 16:0
Author Response
Dear Reviewer #1,
Thank you for handling our manuscript.
We are grateful to you for the careful analysis of our data, useful comments and valuable remarks. Following your very useful recommendations, we have revised the Discussion and Material and Methods sections and Table 4 in the manuscript accordingly.
We used "Track Changes" function to clear our editing in the manuscript.
Please find below our point-to-point responses to your remarks and comments.
Thank you very much for your consideration.
With the best regards,
On behalf of all coauthors,
Dr. Elena Marusich, PhD

Reviewer 2 Report
Dear Authors, your manuscript concerning the antimicrobial activity of fatty acids-enriched fractions of Hermetia illucens is very interesting and new. The introduction, results and discussion are well detailed, but figure (2 and 4) are unclear and the captions are unclear. Please rewrite the captions and ameliorate the quality of the figures also in Supplementary data S1.
At row 68 separe hydrophyla from is; control the spacing between words all along the text.
Author Response
Dear Reviewer #2,
Thank you for handling our manuscript.
We are grateful to you for the careful analysis of our data, useful comments and valuable remarks. Following your very useful recommendations, we have revised Figure 2 and 4 in the Results section and Figures in Supplementary data S1 section in the manuscript accordingly.
We used "Track Changes" function to clear our editing in the manuscript.
Please find below our point-to-point responses to your remarks and comments.
Thank you very much for your consideration.
With the best regards,
On behalf of all coauthors,
Dr. Elena Marusich, PhD

Round 2
Reviewer 1 Report
The article is really more interesting now and it show a high value to read for readers.